# Silver Chloride/Ferricyanide-Based Quasi-Reference Electrode for Potentiometric Sensing Applications

**Khiena Z. Brainina** [1,2,*], **Aleksey V. Tarasov** [1] **and Marina B. Vidrevich** [1]

1   Research and Innovative Center of Sensor Technology, Ural State University of Economics, 8 Marta St. 62, Yekaterinburg 620144, Russia; tarasov_a.v@bk.ru (A.V.T.); mbv@usue.ru (M.B.V.)
2   Department of Analytical Chemistry, Institute of Chemical Engineering, Ural Federal University, Mira St. 19, Yekaterinburg 620002, Russia
*   Correspondence: baz@usue.ru

**Abstract:** Processes' occurring at the Ag/AgCl/Cl$^-$, ([Fe(CN)$_6$]$^{3-/4-}$) ions interface study results are presented. Conditions are selected for the mixed salts' precipitate formation on the silver surface. It has been shown that the potential of a silver screen-printed electrode (AgSPE) coated with a mixed precipitate containing silver chloride/ferricyanide is stable in the presence of [Fe(CN)$_6$]$^{3-/4-}$. The electrode can serve as a quasi-reference electrode (QRE) in electrochemical measurements in media containing ions [Fe(CN)$_6$]$^{3-/4-}$. The electrode is formed during polarization of AgSPE (0.325 V vs. Ag/AgCl/KCl, 3.5 M) in a solution containing chloride- and ferri/ferrocyanides ions. The results of the obtained QRE study by potentiometry, scanning electron microscopy and cyclic voltammetry are presented. The proposed QRE was used in a sensor system to evaluate the antioxidant activity (AOA) of solutions by hybrid potentiometric method (HPM). The results of AOA assessment of fruit juices and biofluids obtained using new QRE and commercial Ag/AgCl RE with separated spaces do not differ.

**Keywords:** quasi-reference electrode; silver screen-printed electrode; electrodeposition; redox couple K$_3$[Fe(CN)$_6$]/K$_4$[Fe(CN)$_6$]; hybrid potentiometric method; antioxidant activity

## 1. Introduction

Processes occurring at the interface silver/solution containing [Fe(CN)$_6$]$^{3-/4-}$, and creating sensor systems capable of functioning in the presence of these ions are of interest because of the following reasons:

- [Fe(CN)$_6$]$^{3-/4-}$ serves as a mediator system in assessing the antioxidant/oxidant activity (AOA/OA) of solutions in hybrid potentiometric (HPM) [1–5] and chronoamperometric (CA) [5–7] methods;
- [Fe(CN)$_6$]$^{3-/4-}$ serves as a mediator system in assessing AOA/OA of solid-phase objects in contact hybrid potentiometric method (CHPM) [8–11];
- [Fe(CN)$_6$]$^{3-/4-}$ is used as a redox probe to study processes in cyclic voltammetry [12–16] and electrochemical impedance spectroscopy [15–17];
- [Fe(CN)$_6$]$^{3-/4-}$ is used in bio- [18,19], immuno- [20] and aptasensors [21].

AOA/OA assessment by HPM [1–5], CA [5–7] and CHPM [8–11] is of great importance in biomedical analysis, as it is a source of information on oxidative stress and health status [22,23]. Evaluation of AOA of food products [3,4,7,11], pharmaceuticals [4], dietary supplements [4] and cosmetics [8] allows to assess their quality, and to determine AOA/OA of biological media [1,2,5,6,8–10] to track the impact of various factors on human health and to evaluate, for example, the effectiveness of the therapy. In this regard, the development of sensors and sensor systems for AOA/OA monitoring is a very urgent and important task.

Sensor systems, as a rule, consist of two or three electrodes, one of which is always a reference electrode (RE). The correctness of measurements is determined by the stability of its potential. A classic and most commonly used RE is a silver/silver chloride electrode [24–28].

Ag/AgCl quasi-reference electrodes (QREs) are described in [29–32]. With all the convenience and functionality of Ag/AgCl QREs, their serious drawback is the non-stability of Ag/AgCl measuring surface from the environment in which it is located. In media containing ions that form sparingly soluble compounds with silver, potential of such electrodes changes uncontrollably [27]. If in voltammetric and amperometric methods, an admissible uncertainty in the RE potential is allowed, more stringent requirements are imposed on the stability of the RE potential in potentiometry [27,28,33]. The situation is more complicated in those cases in which concentration of components (for example, $[Fe(CN)_6]^{3-/4-}$) changes during analysis. This fact causes the challenge of applying Ag/AgCl QREs in sensor systems.

For the described solid-state reference electrodes (SSREs), a number of attempts have been made to protect the Ag/AgCl measuring surface from the impact of environment. To prevent AgCl dissolution, the thickness of the Ag and AgCl layers is increased [34]; protective coatings of polyurethane [35,36], Nafion [36] or graphene oxide [37] are used; an AgCl layer of a special structure (horizontally from the periphery of the template) in combination with an external protective coating of polyimide [38] is formed. In other cases, an external electrolyte-doped polymer is used [39–42], which has a dual function: it provides a relatively constant concentration of chloride anions and creates a diffusion barrier that slows down the dissolution of AgCl. A significant part of Ag/AgCl SSREs' design includes combination of an intermediate electrolyte-doped polymer material and an external protective coating [43–50]. Encapsulation of Ag/AgCl/Cl$^-$ into strong hardening polymers such as polyvinyl chloride [51,52] or poly (n-butyl acrylate) [53–55] is described in recent studies. SSREs described in [26,31–33,35] keep the potential constant in a solution containing $[Fe(CN)_6]^{3-/4-}$, but protective polymer layers, in some cases, create a barrier for the penetration of substances generating the analytical signal to the measuring surface. Additionally, the complexity of such electrodes' manufacturing makes their mass use unlikely. Thus, creation of RE or QRE for sensor systems operating in the presence of $[Fe(CN)_6]^{3-/4-}$ still faces the problem of the reference electrode selection.

The aim of this work is to study the processes occurring at the interface silver/solution containing Cl$^-$ and $[Fe(CN)_6]^{3-/4-}$; searching the information needed for creation of a new QRE that maintains a constant potential in the presence of substances interacting with silver; and demonstration of the analytical applicability of this QRE as part of a sensor system designed to analyze solution's AOA by HPM.

## 2. Materials and Methods

### 2.1. Chemicals and Materials

The following chemicals were used: $K_3[Fe(CN)_6]$, KCl, $Na_2HPO_4 \times 12H_2O$ (CJSC Vecton, St. Petersburg, Russia); $KNO_3$ (AO Reachim Ltd., Moscow, Russia); $K_4[Fe(CN)_6] \times 3H_2O$ (CJSC Kupavnareaktiv, Staraya Kupavna, Russia); $KH_2PO_4$ (NevaReaktiv Ltd., St. Petersburg, Russia). These reagents were chemically pure. Other chemicals were: $H_2SO_4$ 0.1 M standard solution (Ural Chemical Products Plant Ltd., Verhnyaya Pyshma, Russia), HCl 0.1 M standard solution (Ekroshim Ltd., St. Petersburg, Russia), acetone pure for analysis (Component-reaktive Ltd., Moscow, Russia), ethanol 95% (Konstanta-farm M Ltd., Moscow, Russia), $FeCl_3 \times 6H_2O$ 97–102% (AppliChem GmbH – An ITW Company, Darmstadt, Germany). The solvent used was deionized (DI) water with a resistivity 18 M$\Omega \times$ cm.

The following materials were used: 0.5 mm thick plate made of K–96 ceramic (96% $Al_2O_3$, 2.36% $SiO_2$, 0.56% CaO and 1.08% MgO) from JSC South Ural Radioceramics Plant (YUzhnouralsk, Russia), conductive silver paste PP–17C from RPE Delta-Pastes Ltd. (Zelenograd, Russia) and Cementit universal from Merz + Benteli AG (Niederwangen, Switzerland).

## 2.2. Instruments and Devices

For potentiometric measurements, pH/ions meters TA–ION (RPE Tomanalyt Ltd., Tomsk, Russia) were used. Voltammetric measurements were performed on an IVA–5 inversion voltammetric analyzer (RPIE Iva Ltd., Yekaterinburg, Russia). Surface morphology of the electrodes was studied using a JSM–6490LV electron scanning microscope (JEOL Ltd., Tokyo, Japan). Polyethylene terephthalate tubes (Chengdu Puth Medical Plastics Packaging Co., Ltd., Chengdu, China) containing a blood coagulation activator ($SiO_2$) were used for blood collection. Whole blood serum was obtained using a CM–6M centrifuge (SIA ELMI, Riga, Latvia). Silver screen-printed electrodes (AgSPEs) were manufactured using a DEK–248 screen-printer (ASM Assembly Systems Weymouth Ltd., Weymouth, UK). DI water with a resistivity 18 MΩ × cm was obtained on an Akvalab UVOI–MF–1812 installation (JSC RPC Mediana-filter, Moscow, Russia).

## 2.3. Electrodes

A platinum electrode (disk diameter 3 mm) in a polyether ether ketone case type 6.1204.310 (Metrohm AG, Switzerland) was used as a control indicator electrode (Pt). Before starting work, Pt was polished using aluminum oxide powder (first used, a grain size of 0.3 μm, then 0.05 μm) deposited on a polishing cloth, and additionally subjected to a cyclic polarization in the potential range from –0.2 to 1.5 V and scanning rate 0.1 V/s in 0.1 M $H_2SO_4$ solution until a stable cyclic voltammogram was obtained [56]. After each regeneration step, Pt was washed with DI water. Control reference electrode was a silver/silver chloride electrode Ag/AgCl/KCl (3.5 M) type 6.0728.040 completed with an electrolyte vessel type 6.1245.010 (Metrohm AG, Switzerland). A silver/silver chloride electrode Ag/AgCl/KCl (3.5 M) type EVL–1M3.1 (JSC Gomel Plant of Measuring Devices, Belorussia) was used in the measurements, unless otherwise indicated. Potential of the EVL–1M3.1 at 25 °C is 0.204 ± 0.003 V relative to SHE [57]. Potential of the EVL–1M3.1 was checked daily in relation to the reference electrode (a deviation of ± 3 mV was considered acceptable). Glassy carbon rod GC–2000 3 mm/100 mm (JSC RI Grafit, Russia) was used as an auxiliary electrode in voltammetric measurements.

AgSPE was prepared by applying two layers of silver paste on a ceramic plate washed with acetone, ethanol and deionized water. Each layer of paste was hardened as follows: heated to 850 °C for 30 min, cooled to 150 °C for 30 min, dried at 150 °C for 15 min and cooled to room temperature. Double paste application provided an average silver layer thickness of 30 μm. The plate was cut and electrodes 5 × 40 mm in size were obtained. The middle part of AgSPE separating the working and contact zones was isolated with a Cementit–acetone mixture in a ratio of 1:5 by v/v, so that the electrode working area was 20 $mm^2$ (5×4 mm). AgSPEs were modified by forming a precipitate on its surface by electrolysis of a solution containing $[Fe(CN)_6]^{3-/4-}$ under various conditions (see Section 3.2.).

## 2.4. Auxiliary Solution

To modify the AgSPEs and carrying out analysis, solution containing 10 mM $K_3[Fe(CN)_6]$, 0.1 mM $K_4[Fe(CN)_6]$, 1 M KCl, 54.5 mM $Na_2HPO_4$ and 12.1 mM $KH_2PO_4$ (pH 7.4) was used. Its composition corresponds to that used to determine AOA of food products [4] and biofluids [1,2,5].

## 2.5. Potentiometric Measurements

For potentiometric studies, the measurement circuit proposed in this work and shown in Figure 1, was applied. The advantage of such a circuit over the traditional two-electrode configuration is the implementation of identical conditions for comparing the electrodes and the reduction of time, since it provides the possibility to obtain the results of two measurements in one experiment.

Electrode stability in the auxiliary solution was evaluated by potentiometric method using measuring circuit shown in Figure 1a. Following parameters were determined:

1.　τ—potential stabilization (establishment) time, s;
2.　$E_1$—steady-state value of potential, mV;

3.  $\Delta E/\Delta t$—drift of the potential, mV/h;
4.  $E_2$—potential value, mV, was obtained when 0.4 mL 0.1 M $K_4[Fe(CN)_6]$ was added to 4 mL of auxiliary solution (the final concentration of $K_4[Fe(CN)_6]$ in the cell is 9.1 mM, which models the upper value of antioxidant content in the sample).
5.  $E_3$—potential value, mV, was established when electrode is returned to the initial solution.

These parameters were evaluated based on the study of four modified AgSPEs (Table 2) and four Ag/AgCl QREs (Table 4). The most stable modified AgSPE was considered as a $QRE_{mix}$.

Solution's (fruit juices and biofluids) AOA were evaluated at room temperature by HPM [1–5]. The measuring circuit, shown in Figure 1b, was used. In the circuit, PtSPE vs. EVL–1M3.1 served as an electrode pair, and PtSPE vs. $QRE_{mix}$ served as a sensor system.

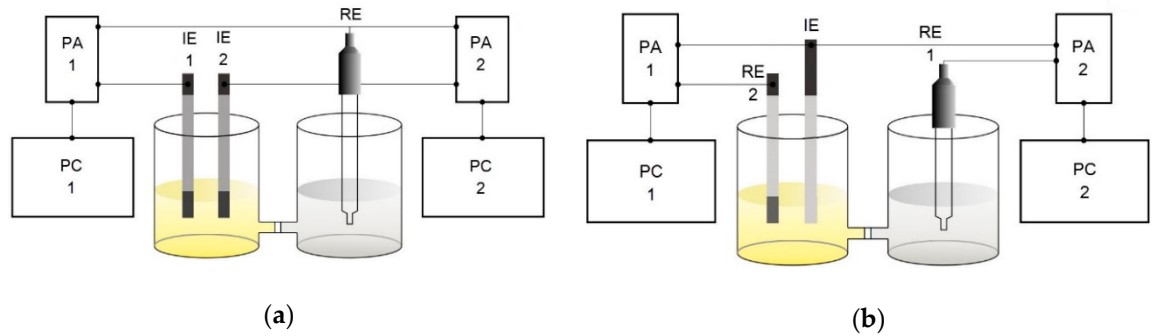

(**a**)            (**b**)

**Figure 1.** Schemes for measuring potential (**a**) of two indicator electrodes (IE 1 and IE 2) with respect to one reference electrode (RE); (**b**) one indicator electrode (IE) relatively to two reference electrodes (RE 1 and RE 2). PA 1 and PA 2: potentiometric analyzers; PC 1 and PC 2: personal computers; IE 1 and IE 2: modified AgSPEs; RE = RE 1: EVL – 1M3.1; RE 2: $QRE_{mix}$.

### 2.6. Voltammetric and Scanning Electron Microscopy Measurements

Normal three-electron cell was used, consisting of AgSPE or Pt, EVL–1M3.1 or $QRE_{mix}$ and GC–2000 as working electrode, reference electrode, and auxiliary electrode. Potential scanning rate was 0.05 V/s. Scanning electron microscopy (SEM) measurements were performed at 20 kV in vacuum. Data were used for electrodes surface morphology characterization (see Section 3.4).

### 2.7. Potentiometric Sensor System Assembly

Platinum screen-printed electrode (PtSPE) was used as an indicator in the evaluation of AOA of fruit juices and biofluids. Applicability of PtSPE in food analysis [4] and biofluids [1,2,5] has been confirmed earlier. After operation in food and biological matrices, PtSPE was regenerated by annealing at a temperature of 750 °C for 1 h [58]. The most stable modified AgSPE (i.e., $QRE_{mix}$) served as a reference electrode. The results obtained are presented in Section 3.6.

### 2.8. Sampling and Sample Preparation

Samples of apple juices of the Dobryj, Rich, J7 brands and fresh apples of Smit and Fuji varieties were purchased at a local supermarket. The juices taken from the packages opened before analysis and freshly squeezed juices were examined. Saliva was taken into a plastic container 10 min before analysis, while the respondent refrained from eating, drinking, smoking and brushing his teeth for at least an hour. Blood was collected by venipuncture at the bend of the elbow joint into a polyethylene terephthalate tube containing a blood coagulation activator $(SiO_2)$. To obtain serum, whole blood samples were centrifuged at 3500 rpm for 15 min. The resulting blood serum was frozen and stored at −18 °C. The ejaculate was taken into a plastic container by natural masturbation after 2–3 days of abstinence. The selected ejaculate samples were kept for 40 min at room temperature, and then they

were frozen and stored at −18 °C. Before analysis, serum and ejaculate samples were thawed for 40 min at room temperature.

### 2.9. Statistical Analysis

All measurements were repeated 4 times. Statistical analysis was performed in Microsoft Excel 2010 with an accepted significance level of $\alpha = 0.05$. The data are presented as $X \pm \Delta X$, where $X$ is the average value, $\Delta X$ is the standard deviation. Validation of the results of the evaluation of AOA solutions obtained on the developed reference electrode ($QRE_{mix}$) was performed in relation to the results obtained on a commercial reference electrode (EVL–1M3.1), based on F- and t-tests.

## 3. Results and Discussion

### 3.1. Study of Redox Processes Occurring on Polarized AgSPE

Figure 2 shows cyclic voltammograms describing the redox processes on AgSPE in solutions of KCl, $K_4[Fe(CN)_6]$, $K_3[Fe(CN)_6]$, $Na_2HPO_4$ and $KH_2PO_4$, which are components of the solution used. Cyclic voltammograms for Pt are also shown for comparison.

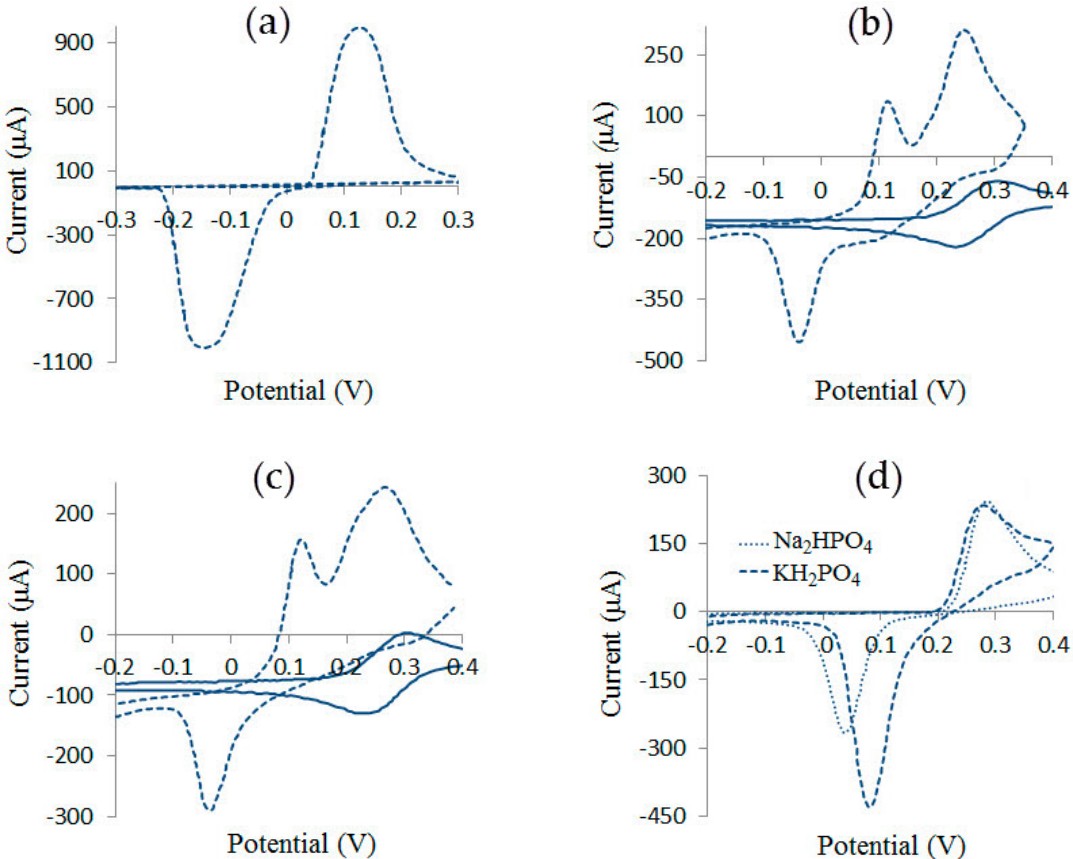

**Figure 2.** Cyclic voltammograms recorded on AgSPE (dashed lines, first cycle) and Pt (solid lines) in solutions of (**a**) 1 M KCl, (**b**) 0.005 M $K_4[Fe(CN)_6]$ and 1 M $KNO_3$, (**c**) 0.005 M $K_3[Fe(CN)_6]$ and 1 M $KNO_3$, (**d**) 0.005 M $Na_2HPO_4$ with 1 M $KNO_3$ and 0.005 M $KH_2PO_4$ with 1 M $KNO_3$. Scanning rate 0.05 V/s.

Taking into account the low solubility of silver salts (Table 1), it should be concluded that the currents in cyclic voltammograms (Figure 2) are due to the following processes:

- Figure 2a: $Ag^0_{(s)} + Cl^-_{(aq.)} \rightarrow AgCl_{(s)} + e^-$ at E > 0.05 V [31,32];

- Figure 2b: $4Ag^0_{(s)} + [Fe(CN)_6]^{4-}_{(aq.)} \rightarrow Ag_4[Fe(CN)_6]_{(s)} + 4e^-$ (first peak at 0.05 V < E< 0.15 V) and $[Fe(CN)_6]^{4-}_{(s)} \rightarrow [Fe(CN)_6]^{3-}_{(s)} + e^-$ (second peak at E> 0.15 V), which is associated with a similar process for an indifferent Pt $[Fe(CN)_6]^{4-}_{(aq.)} \rightarrow [Fe(CN)_6]^{3-}_{(aq.)} + e^-$;
- Figure 2c: the same processes as in Figure 2b ($[Fe(CN)_6]^{4-}$ ions appear as a result of $[Fe(CN)_6]^{3-}$ ions reduction in potential scanning process);
- Figure 2d: $3Ag^0_{(s)} + PO_4^{3-}_{(aq.)} \rightarrow Ag_3PO_{4(s)} + 3e^-$ at E > 0.2 V.

**Table 1.** The equilibrium concentration of silver ions over solutions of sparingly soluble compound.

| Ion Concentration, M [1] | Sparingly Soluble Compound, its Solubility Product [59,60] [2] | Equilibrium Concentration of Silver Ions, M |
|---|---|---|
| $Cl^-$, 1 | AgCl, $1.8 \times 10^{-10}$ | $1.8 \times 10^{-10}$ |
| $[Fe(CN)_6]^{4-}$, $1 \times 10^{-4}$ | $Ag_4[Fe(CN)_6]$, $1.5 \times 10^{-41}$ | $6.2 \times 10^{-10}$ |
| $[Fe(CN)_6]^{3-}$, $1 \times 10^{-2}$ | $Ag_3[Fe(CN)_6]$, $9.8 \times 10^{-26}$ | $2.1 \times 10^{-8}$ |
| $PO_4^{3-}$, $6.6 \times 10^{-2}$ | $Ag_3PO_4$, $8.9 \times 10^{-17}$ | $1.1 \times 10^{-5}$ |

[1] At salt dissociation degree 100%; [2] At 25 °C.

In a mixture of components, redox processes considered can take place in parallel or as competing ones. Composition of the precipitate formed on AgSPE depends on concentration of anions in the solution, solubility product of silver compounds, and electrode potential. In the potential range 0.05 V < E < 0.15 V, AgCl and $Ag_4[Fe(CN)_6]$ are formed together; at 0.15 V < E< 0.2 V, AgCl and $Ag_3[Fe(CN)_6]$ are formed; and at E > 0.2 V formation of three sparingly soluble compounds AgCl, $Ag_3[Fe(CN)_6]$ and $Ag_3PO_4$ is possible. Since, predominantly, those poorly soluble compounds are formed, in equilibrium with which concentration of silver ions is minimal (Table 1), formation of $Ag_3PO_4$ under these conditions can be neglected.

### 3.2. Selection of Surface Formation Conditions of QRE_mix

Table 2 presents modification conditions of AgSPE. AgSPE polarization modes are selected according to the cyclic curves shown in Figure 2.

**Table 2.** AgSPE modification conditions in the solution containing $[Fe(CN)_6]^{3-/4-}$ at room temperature.

| Mode | Mixing | Potential, V | Modification Time, s |
|---|---|---|---|
| Open circuit | no/yes | – | 600; 1800; 3600; 5400; 7200 |
| Potentiostatic | yes | 0.1; 0.15; 0.2; 0.25; 0.3; 0.325; 0.35; 0.4 | 15; 30; 60; 120 |
| Potentiodynamic [1] | yes | 0.05–0.15; 0.15–0.2; 0.2–0.35 | 120 |

[1] 0.05 V/s.

Figure 3 shows dependences of the modified AgSPE potential stabilization time ($\tau$) on modification duration. As can be seen from Figure 3a, potential of an open-chain modified AgSPE with solution stirring stabilizes faster than without solution stirring. Minimum $\tau$ values for open-chain modified AgSPEs are observed at modification duration of 7200 s (2 h). Parameters $\tau$, $\Delta E/\Delta t$, $E_1$, $E_2$, $E_3$ for AgSPE modified in an open circuit without solution stirring are poorly reproducible. Potential stabilization time of AgSPE modified in the potentiostatic mode decreases with increasing of modification duration (Figure 3b). Minimum $\tau$ values are observed after modifying electrodes for 120 s. Results in which standard deviation of one or more parameters $E_1$, $E_2$, and $E_3$ exceeds 3 mV are excluded from further consideration, since they do not satisfy necessary conditions. Results in which standard deviation of the parameters $E_1$, $E_2$ and $E_3$ is less than 3 mV are shown in Table 3.

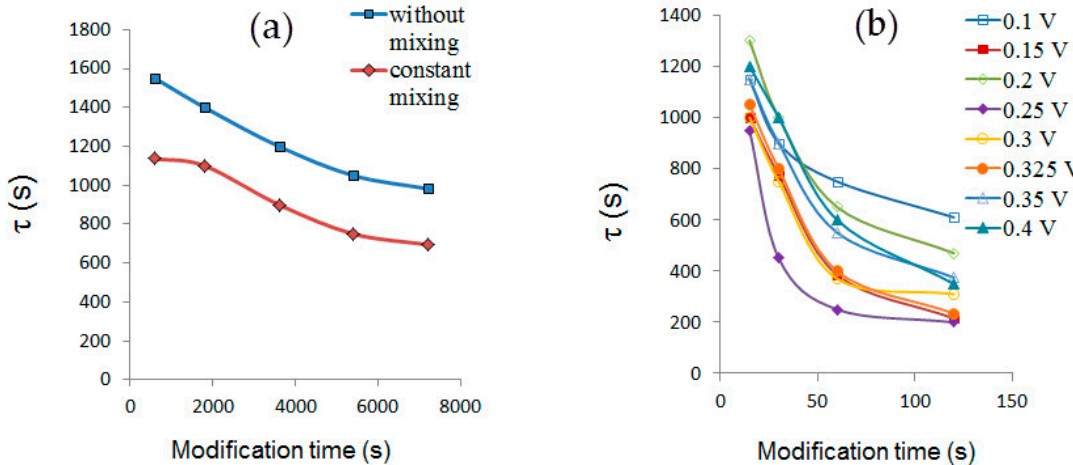

**Figure 3.** AgSPE potential stabilization time (average values for n = 4 shown) versus open circuit modification time (**a**) and potentiostatic mode (**b**).

**Table 3.** Influence of AgSPE modification conditions in $[Fe(CN)_6]^{3-/4-}$ containing solution on the values of parameters characterizing electrode properties.

| Modification Conditions | | $\tau$, s | $\Delta E/\Delta t$, mV/h | $E_1$, mV | $E_2$, mV | $E_3$, mV |
|---|---|---|---|---|---|---|
| Mode | Conditions | | | | | |
| OC [1] | 7200 s | 695 ± 413 | 2.0 ± 0.6 | 66 ± 1 | 63 ± 1 | 61 ± 2 |
| | 0.3 V, 120 s | 310 ± 155 | 0.5 ± 0.3 | 38 ± 1 | 36 ± 1 | 40 ± 1 |
| PsM [2] | 0.325 V, 120 s | 235 ±74 | 0.4 ± 0.2 | 45 ± 1 | 45 ± 1 | 44 ± 1 |
| | 0.35 V, 120 s | 375 ±176 | 0.6 ± 0.4 | 54 ± 0 | 57 ± 1 | 54 ± 0 |

[1] Open circuit; [2] Potentiostatic mode.

It follows from Table 3 that optimal conditions are: potentiostatic polarization mode at 0.325 V for 120 s. Further, the electrode formed under these conditions was considered as a new QRE and defined as $QRE_{mix}$, which indicated presence of a mixed precipitate on its surface. In this case, precipitate consisted mainly of AgCl and $Ag_3[Fe(CN)_6]$ (see Section 3.1).

### 3.3. Comparison of $QRE_{mix}$ and Ag/AgCl QRE

Conditions of Ag/AgCl QRE formation are presented in Table 4. The results of a comparative study of Ag/AgCl QRE and $QRE_{mix}$ in solution containing $[Fe(CN)_6]^{3-/4-}$ stability are shown in Figure 4. It can be seen from Figure 4 that $QRE_{mix}$ is the most stable. In this case, parameter's ($\tau$, $\Delta E/\Delta t$, $E_1$, $E_2$, $E_3$) values that determine operability of the electrode are minimal. $QRE_{mix}$ was stored in air at room temperature without direct sunlight. Parameters preserve normal values for application as a reference electrode for 7 days (Figure 4).

**Table 4.** Ag/AgCl QRE types and their formation conditions.

| Type | AgCl Layer Formation Conditions | Source |
|---|---|---|
| 1 | Ag oxidation in 50 mM $FeCl_3$ for 50 s | [34,37] |
| 2 | Ag oxidation in 0.1 M KCl at 0.5 V for 120 s | [43,44] |
| 3 | Ag oxidation in 0.1 M HCl at 0.145 V (potential 50 mV more positive than open circuit potential) for 120 s | [61] |
| 4 | Ag oxidation in 1 M KCl at 0.325 V for 120 s | [This work] |

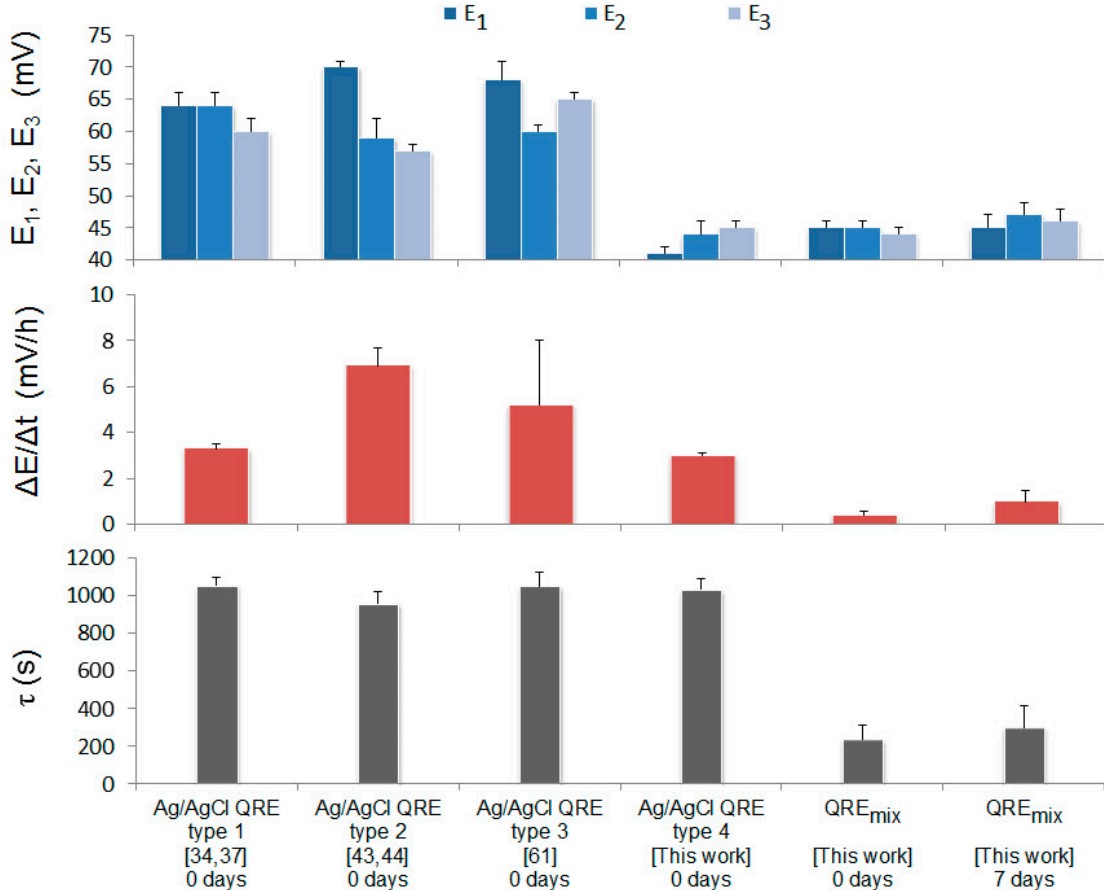

**Figure 4.** Experimental values of the parameters ($\tau$ is potential stabilization time, $\Delta E/\Delta t$ is potential drift and $E_1$, $E_2$, $E_3$ are potentials values) characteristic for $QRE_{mix}$ and Ag/AgCl QREs depending on the electrodes storage duration before measurements. The boundaries indicate upper values of standard deviation for $\alpha = 0.05$ and n = 4.

### 3.4. Electrode Surface Morphology

Figure 5 shows SEM images of AgSPE surface (Figure 5a) and AgSPE's surface, which are formed at a potential of 0.325 V in non-containing and containing $[Fe(CN)_6]^{3-/4-}$ solutions (Figure 5b,f). AgSPE surface is a conglomerate of polydisperse crystals 2–10 µm size (Figure 5a). Ag/AgCl QRE type 4 and $QRE_{mix}$ surface (Figure 5b,c) consists of polydisperse large (0.4–1.8 µm) crystals. Fine crystalline precipitates form on AgSPEs obtained in $[Fe(CN)_6]^{3-/4-}$ solutions (Figure 5d,f). Probably a fine-crystalline precipitate leave out, mainly of $Ag_3[Fe(CN)_6]$, leave out as shown earlier (see Section 3.1), fills pores between large AgCl crystals, which leads to an improvement of $QRE_{mix}$ stability compared to Ag/AgCl QREs.

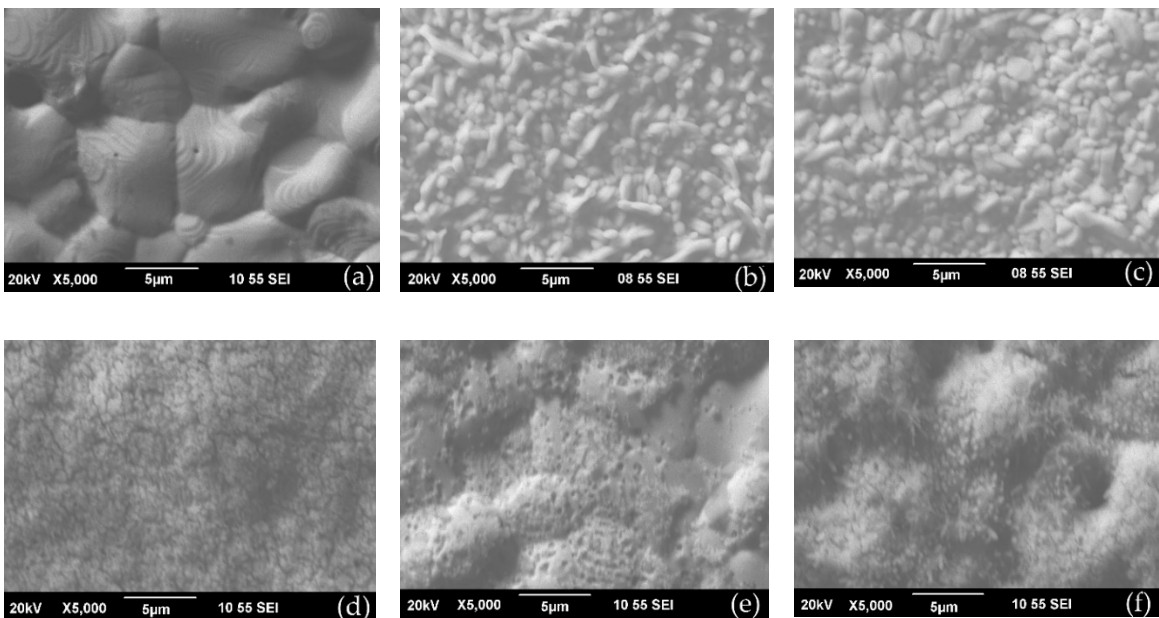

**Figure 5.** SEM images of surfaces: (**a**) AgSPE, (**b**) Ag/AgCl QRE type 4, (**c**) QRE$_{mix}$ and AgSPEs modified in potentiostatic mode (0.325 V, 120 s) in solutions containing (**d**) 5 mM K$_3$[Fe(CN)$_6$], 54.5 mM Na$_2$HPO$_4$, 12.1 mM KH$_2$PO$_4$ and 1 M KNO$_3$, (**e**) 5 mM K$_4$[Fe(CN)$_6$], 54.5 mM Na$_2$HPO$_4$, 12.1 mM KH$_2$PO$_4$ and 1 M KNO$_3$, (**f**) 5 mM K$_3$[Fe(CN)$_6$], 5 mM K$_4$[Fe(CN)$_6$], 54.5 mM Na$_2$HPO$_4$, 12.1 mM KH$_2$PO$_4$ and 1 M KNO$_3$.

### 3.5. QREmix Characterization by Cyclic Voltammetry

Cyclic voltammograms recorded on Pt in a solution containing 1 mM [Fe(CN)6]$^{3-/4-}$ (1:1) and 1 M KCl using EVL–1M3.1 or QREmix as a reference electrode are shown in Figure 6. It is seen that the obtained cyclic voltammograms are almost identical and differ only in a parallel shift along the potential axis by 45 ± 3 mV (n = 4), which corresponds to the potential difference between EVL–1M3.1 and QREmix. This is one more evidence of QREmix stable operation in a solution containing [Fe(CN)6]$^{3-/4-}$. It may contribute to the use of this electrode for other electrochemical applications.

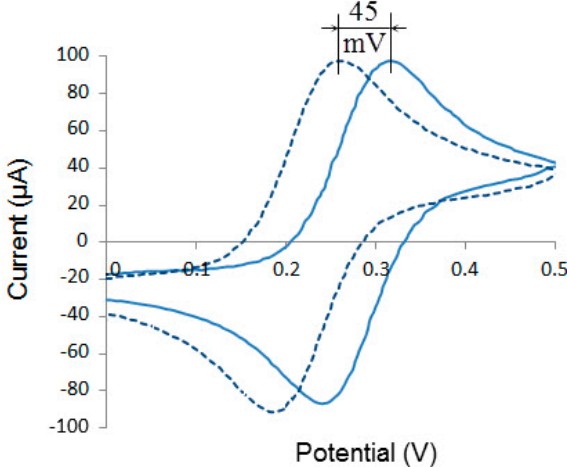

**Figure 6.** Cyclic voltammograms recorded on Pt in a solution containing 1 mM K$_3$[Fe(CN)$_6$], 1 mM K$_4$[Fe(CN)$_6$] and 1 M KCl, using EVL–1M3.1 (solid line) and QRE$_{mix}$ (dashed line) as a reference electrode.

### 3.6. Potentiometric Sensor System in the Determination of AOA of Solutions

The QRE$_{mix}$ proposed was used as part of the sensor system to evaluate AOA of fruit juices and biofluids by HPM in comparison with commercial Ag/AgCl RE. Measurement circuit shown in

Figure 1b (IE = PtSPE, RE 1 = EVL–1M3.1 and RE 2 = $QRE_{mix}$) was used. The results are presented in Table 5. It can be seen from Table 5 that values of F- and t-tests are less than theoretical ones, which proves the same reproducibility and statistically insignificant differences between the results. The data presented indicate correctness of using of $QRE_{mix}$ in the real samples' analysis.

**Table 5.** AOA determination results of solutions (fruit juices and biofluids) obtained using the PtSPE vs. $QRE_{mix}$ and PtSPE vs. EVL–1M3.1 (n = 4).

| Sample | $QRE_{mix}$ | | EVL–1M3.1 | | F [2] | t [3] |
|---|---|---|---|---|---|---|
| | mmol-eq/L | $S_r$ [1] | mmol-eq/L | $S_r$ [1] | | |
| Apple juice Dobryi | 2.05 ± 0.07 | 0.03 | 1.95 ± 0.03 | 0.02 | 4.00 | 2.08 |
| Apple juice Rich | 3.20 ± 0.06 | 0.02 | 3.12 ± 0.04 | 0.01 | 1.78 | 1.96 |
| Apple juice J7 | 3.56 ± 0.06 | 0.02 | 3.48 ± 0.03 | 0.01 | 4.00 | 2.19 |
| Apple fresh Smit | 5.30 ± 0.14 | 0.03 | 5.02 ± 0.17 | 0.03 | 1.44 | 2.19 |
| Apple fresh Fuji | 6.22 ± 0.18 | 0.03 | 6.07 ± 0.09 | 0.01 | 3.70 | 1.30 |
| Saliva | 1.00 ±0.07 | 0.07 | 0.84 ± 0.03 | 0.03 | 6.25 | 0.48 |
| Blood serum | 1.27 ± 0.03 | 0.03 | 1.07 ± 0.03 | 0.03 | 1.00 | 1.76 |
| Semen (ejaculate) | 1.50 ± 0.14 | 0.09 | 1.35 ± 0.07 | 0.05 | 4.00 | 0.90 |

[1] Relative standard deviation; [2] Fisher's test ($F_{teor.}$ = 9.28 for $f_1 = n_1 - 1 = 3$, $f_2 = n_2 - 1 = 3$ and $\alpha = 0.05$); [3] Student's t test ($t_{teor.}$ = 2.45 for $f = n_1 + n_2 - 2 = 6$ and $\alpha = 0.05$).

## 4. Conclusions

The results of a study of the processes occurring at the electrode/multianionic interface, some of which form sparingly soluble or complex compounds with the electrode material, are of general interest. The data obtained in this work make a definite contribution to this area: they justify research paths and provide the information needed for creating QREs for sensor systems operating in the presence of interfering ions, $[Fe(CN)_6]^{3-/4-}$ in particular. The latter allowed us to solve a very important problem—to develop sensory systems for monitoring AOA/OA of various objects, including vital biological ones. The QRE proposed consists of an Ag screen-printed substrate electrochemically coated with a mixed precipitate containing silver chloride/ferricyanide. The sediment was studied by potentiometry, scanning electron microscopy and cyclic voltammetry. A new electrochemical scheme is proposed and used, which allows comparing electrodes in one measurement. This approach increases the accuracy of the research results and reduces time required to obtain them. Performance of the sensor system with the new QRE is illustrated by its application for determination of AOA of fruit juices and body liquids. The results obtained have good analytical characteristics, which allows us to predict the widespread use of QRE described in this paper. Additional strategies aimed at improving the stability of the developed QRE may include, but are not limited to, the use of protective coatings. The approaches and methods described may be useful for creating QREs working in the presence of other interfering substances.

**Author Contributions:** Conceptualization, K.Z.B.; validation, A.V.T.; formal analysis, A.V.T.; investigation, A.V.T.; resources, A.V.T.; writing—original draft preparation, K.Z.B., A.V.T. and M.B.V.; writing—review and editing, K.Z.B. and M.B.V.; visualization, A.V.T.; funding acquisition, A.V.T. All authors have read and agreed to the published version of the manuscript.

**Funding:** The reported study was funded by RFBR according to the research project № 18-33-00215.

**Acknowledgments:** The authors are grateful to the Laboratory of Structural Methods of Analysis and the Properties of Materials and Nanomaterials in the Ural Federal University named after the first President of Russia, B.N. Yeltsin, for conducting electrodes SEM.

**Conflicts of Interest:** The authors declare no conflict of interest.

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
