# Peer review of "Silver Chloride/Ferricyanide-Based Quasi-Reference Electrode for Potentiometric Sensing Applications"

_chemosensors, doi:10.3390/chemosensors8010015_

Round 1
Reviewer 1 Report
Please see the attached review comments.

Author Response
Point 1: English language and style are fine/minor spell check required.
Response 1: Spelling is checked.
Point 2: The title of the work does not reflect the manuscript content. The work focused mainly on developing the quasi-reference electrode and its characterization, and just one section on AOA monitoring. Therefore, I recommend the title to be changed to something like ''Silver Ferri/Ferrocyanides-based Quasi-Reference Electrode for Potentiometric Sensing Applications''.
Response 2: Thanks. It’s done.
Point 3: Apart from ΔE/Δt, the characterization of the quasi-reference electrode (QRE) does not provide any temporal data (e.g., open circuit potential vs. time) with respect to a commercial Ag/AgCl glass electrode. This type of results would be useful to show the stability and long-term drift behavior of the proposed QRE. Such type of experiments should be done for at least few days to and be compared with other reported literatures.
Response 3: Investigations were carried out. Potential is stable for a long time. We do not think that the results are worth being included into the text taking into account 2 issues: (i) electrode is intended for use as a disposable one and (ii) article too overloaded with experimental data.
Point 4: Section 2.4, 2.6, 2.7, 2.8, and in few other instances, the paragraphs contain one sentence only. I standard paragraph should contain at least 4/5 sentences. The authors may combine small sections one paragraph or elaborate them each.
Response 4: Thanks. It’s done.
Point 5: Some section titles are loo long, for example, section 3.1 and 3.2.
Response 5:
3.1. Study of Redox Processes Occurring on Polarized AgSPE
3.2. Selection of Surface Formation Conditions of QREmix
Point 6: Conclusions should normally be one paragraph. A lot of paragraphs in the conclusions have only one sentence. I encourage to follow journal standards in writing conclusions.
Response 6: It’s improved.

Reviewer 2 Report
The paper can be published after minor revision reflelecting comments inserted as yellow notes into pdf of submitted manuscript. This pdf was emailed separately to Betsy Wang [email protected] and it should be forwarded to authors

Author Response
Point 1: Moderate English changes required.
Response 1: Spelling is checked.
Points 2–9: Reviewer comments and Authors responses are presented in the table.
Point |
Response |
|
Line (earlier/now) |
Comment |
|
11/ 13-14 |
The results of studying processes occurring at the Ag/AgCl/Cl–, ([Fe(CN)6]3–/4–) ions interface are presented. |
Processes’ occurring at the Ag/AgCl/Cl–, ([Fe(CN)6]3–/4–) ions interface study results are presented. |
13/ 14-15 |
Conditions for the mixed salts precipitate formation on the silver surface are selected. |
Conditions are selected for the mixed salts’ precipitate formation on the silver surface. |
16-17/ 19 |
The electrode is formed upon polarization of silver (0.325 V vs. Ag/AgCl/KCl, 3.5 M) in a solution containing chloride and ferri/ferrocyanides ions. |
The electrode is formed during polarization of AgSPE (0.325 V vs. Ag/AgCl/KCl, 3.5 M) in a solution containing chloride- and ferri-/ferrocyanides ions. |
27/ 30-32 |
The interest in studying the processes occurring at the silver / solution interface containing [Fe(CN)6]3–/4–, and creating sensor systems capable of functioning in the presence of these ions is caused by the following reasons: |
Processes occurring at the interface silver / solution containing [Fe(CN)6]3–/4–, and creating sensor systems capable of functioning in the presence of these ions are of interest because of the following reasons: |
49-50/ 52 |
With all the convenience and functionality of Ag/AgCl QREs, their serious drawback is the insecurity of the Ag/AgCl measuring surface from the environment in which it is located. |
With all the convenience and functionality of Ag/AgCl QREs, their serious drawback is the non-stability of Ag/AgCl measuring surface from the environment in which it is located. |
52-53/ 55-56 |
If in voltammetric and amperometric methods there is an admissible uncertainty in the RE potential is allowed, then more stringent requirements are imposed on the stability of the RE potential in potentiometry [27,28,33]. |
If in voltammetric and amperometric methods an admissible uncertainty in the RE potential is allowed, more stringent requirements are imposed on the stability of the RE potential in potentiometry [27,28,33]. |
71-72/ 75-76 |
Thus, creation of RE or QRE for sensor systems operating in the presence of [Fe(CN)6]3–/4– still faces the reference electrode problem. |
Thus, creation of RE or QRE for sensor systems operating in the presence of [Fe(CN)6]3–/4– still faces the problem of the reference electrode selection. |
182-183/ 189-190 |
Validation of the results of the evaluation of AOA solutions obtained using the developed reference electrode (QREmix) was performed in relatively to the results obtained using a commercial reference electrode (EVL–1M3.1), based on F- and t-tests. |
Validation of the results of the evaluation of AOA solutions obtained on the developed reference electrode (QREmix) was performed in relation to the results obtained on a commercial reference electrode (EVL–1M3.1), based on F- and t-tests. |

Reviewer 3 Report
The manuscript describes the use of silver screen-printed electrode (AgSPE), coated with a mixed precipitate containing silver ferri/ferrocyanides and chloride ions, as a quasi reference electrode in electrochemical measurements in media containing interfering ions, especially [Fe(CN)6]3–/4– . Electrode stability was evaluated by potentiometric method and voltammetry and the electrode surface morfology was studied with SEM. The electrode was used as a sensor system to evaluate antioxidant activity in fruit juices and biofluids. The results were in good agreement with an Ag/AgCl reference electrode.
I find the manuscript well written and suggest its publication in a present form.
Author Response
Point 1: English language and style are fine/minor spell check required.
Response 1: Thanks. It’s done.
Point 2: I find the manuscript well written and suggest its publication in a present form.
Response 2: Thank you very much.
